# From Flourish to Nourish: Cultivating Soil Health for Sustainable Floriculture

**DOI:** 10.3390/plants13213055

**Published:** 2024-10-31

**Authors:** Peihua Zhang, Jie Zhou, Di He, Yiran Yang, Zhenhong Lu, Chunmei Yang, Dongdong Zhang, Fan Li, Jihua Wang

**Affiliations:** 1Key Laboratory for Flower Breeding of Yunnan Province, Floriculture Research Institute, Yunnan Academy of Agricultural Sciences, National Engineering Research Center for Ornamental Horticulture, Kunming 650200, China; luzhenhong@yaas.org.cn (Z.L.); yangchunmei@yaas.org.cn (C.Y.); 2Yunnan Seed Laboratory, Kunming 650200, China; 3International Agricultural Research Institute, Yunnan Academy of Agricultural Sciences, Kunming 650200, China; 4School of Agriculture, Yunnan University, Kunming 650200, China; 12022228116@mail.ynu.edu.cn (J.Z.); hedi1@stu.ynu.edu.cn (D.H.); yangyiran@stu.ynu.edu.cn (Y.Y.); 5Institute of Urban Agriculture, Chinese Academy of Agricultural Sciences, Chengdu 610213, China; zhangdongdong@caas.cn

**Keywords:** floriculture sustainability, soil health, sustainable soil management, floral wastes

## Abstract

Despite its rapid growth and economic success, the sustainability of the floriculture industry as it is presently conducted is debatable, due to the huge environmental impacts it initiates and incurs. Achieving sustainability requires joint efforts from all stakeholders, a fact that is often neglected in discussions that frequently focus upon economically driven management concerns. This review attempts to raise awareness and collective responsibility among the key practitioners in floriculture by discussing its sustainability in the context of soil health, as soil is the foundation of agriculture systems. Major challenges posed to soil health arise from soil acidification and salinization stimulated by the abusive use of fertilizers. The poisoning of soil biota by pesticide residues and plastic debris due to the excessive application of pesticides and disposal of plastics is another significant issue and concern. The consequence of continuous cropping obstacles are further elucidated by the concept of plant-soil feedback. Based on these challenges, we propose the adoption and implementation of several sustainable practices including breeding stress-resistant and nutrient-efficient cultivars, making sustainable soil management a goal of floriculture production, and the recycling of plastics to overcome and mitigate the decline in soil health. The problems created by flower waste materials are highlighted and efficient treatment by biochar synthesis is suggested. We acknowledge the complexity of developing and implementing the proposed practices in floriculture as there is limited collaboration among the research and operational communities, and the policymakers. Additional research examining the impacts the floriculture industry has upon soils is needed to develop more sustainable production practices that can help resolve the current threats and to bridge the understanding gap between researchers and stakeholders in floriculture.

## 1. Introduction

Floriculture, consists of cut flowers, potted plants, cut foliage, and bedding/garden plants, is one of the most important commercial trades with striking performances in agriculture [1]. Over 134 countries are active participants in the global trade of floriculture plants and products with developed countries in Europe and North America as the most prominent consumers whereas the Netherlands and developing African, Asian and American countries are the major producers [2]. The global market for floriculture industry is expected to grow from 49.8 billion US dollars in 2022 to 106.1 billion US dollars in 2033 [3]. Of this total, cut flowers’ production (estimated by exported sum) was valued at 105 billion US dollars in 2023 [4]. A total of 735,500 ha land were used in floriculture industry including cut flowers and potted plants as estimated by the international Association of Horticultural Producers [5]. Although world economic growth has declined due to the COVID-19 pandemic, the product values for cut flowers in 2022 has experienced a 21.38% increase, indicating the promising development of this industry [5].

Floriculture products rank the first regarding the profit per unit area when compared to other agricultural products [6]. However, the supply of floriculture products is highly dependent on energy intensive production methods from breeding and cultivation to harvesting and distribution [7]. In the pursuit of flower products with high qualities and yield, floriculture crops are produced under controlled environment or protected structures with substantial supplies of fertilizers, pesticides, growth regulators, greenhouse heating and glazing which mostly derived from nonrenewable and petroleum-based products [8]. Moreover, these crops are commonly grown and packaged in non-degradable plastic films, containers, bags and packs [9]. Hence, the floriculture industry is highly debatable with the huge environmental impacts resulting from the excessive inputs of agrochemicals and energy consumption at each stage within the conventional floriculture production system [10].

Few modern industrial ornamental horticultural enterprises can be recognized as environmentally, socially or economically sustainable, and it is also difficult for growers to adjust and apply sustainable practices due to economic pressure resulting from the increasing competitive marketplace for floriculture products [11,12]. The sustainability of the floriculture industry has already become a prominent research topic in recent years [10,13]. Among these, the global environmental concerns in environmental pollution, depletion of water resources, waste generation, and biodiversity loss due to the indiscriminate usage of peat, fertilizers and pesticides were intensively discussed among researchers [14,15,16,17,18]. The emergence of the Slow Flower Movement also indicates the demand from customers for sustainable alternatives to the conventional floriculture [19]. Although sustainability challenges have been well recognized from the researchers’ and consumers’ perspective, a prevailing weak sustainability position can be identified among floricultural stakeholders (including farmers, companies, and policymakers) due to their economic-oriented thinking [13]. Environmental sustainability is a collective responsibility and concern for all stakeholders including farmers, employers, consumers, government, and policymakers [20]. Thus, bridging the gap in understanding between researchers and stakeholders is indispensable and requires explicit elaborations on the current challenges regarding the sustainability of the floricultural industry.

Soil plays a pivotal role in achieving many sustainable development goals of the United Nations such as food security, life on land, climate action, clean water, and resource efficiency [21]. The importance of soil health in sustaining agricultural systems is well-recognized [22,23]. Nevertheless, the increasing demand from customer and industry for soil-based products leads to an overexploitation of soil resources and increase of soil disturbance [24,25]. Once exterior disturbances are beyond the reach of soil resilience, soil begins to degrade. This has often resulted in soil degradation such as contamination, acidification, salinization and erosion [26,27]. Soil is the foundation of agriculture systems, including floriculture, because of its ecological functions in mechanically supporting of plants, nutrients provision and water supply for plant growth [28]. Among the five main types of cut flowers, four are cultivated in the soil including carnation, lilium, chrysanthemum, and lisianthus. However, modern floriculture systems often establish their production levels on the basis of economic goals instead of the soil capacity to withstand exterior stresses such as the over-application of agrochemicals. With these increasingly frequent disturbances and the slow formation nature of soil (2.5 cm in 150 years), soils as non-renewable resources are easily damaged with their functions impaired [21]. As the quality of floriculture products must be obtained in the cultivation stage, the primary goal of sustainable floriculture requires the protection and enhancement of soil health [8]. Achieving and sustaining soil health requires major adaptations of socio-economic activities, policies, and consumption patterns by increasing engagement among multi-stakeholders, inter-disciplinary and multi-scale operational research and innovation [25]. Solutions for the future sustainable development of floriculture may also embeds in the identified causes for the negative consequences. Thereupon, discussing floriculture sustainability in the context of soil health can resonate among all key stakeholders which will make joint efforts to realize their collective responsibility.

The objectives of this review are to firstly scrutinize the existing threats posed to the soil environment in the floriculture production system. An evaluation is then conducted on the extents of these environmental impacts on the sustainability of floriculture industry from aspects of soil physiochemical properties and biological activities. Based on these threats and resulting negative impacts, we discuss the potential benefits of including multiple agronomic incentives and strategies to improve the current situation to enhance the soil health. The pollution of floral wastes is highlighted and efficient treatment by biochar synthesis is advocated supported by latest findings.

## 2. Green but Not “Green” Floriculture: Potential Threats to the Soil Ecosystems

Due to the indiscriminate usage of fertilizers, pesticides and plastics along with the poor treatment of flower wastes, soils in the floriculture agroecosystem face many threats as summarized in Figure 1.

### 2.1. Soil Biota Poisoned by Toxic Pesticide Residues

Flower is the most pesticide-intensive crop due to the frequent outbreak of pests and pathogens in closed greenhouse environments [16]. The application of pesticides in cut flowers can reach up to over 22.4 kg per ha which is 33 times higher than wheat, and 9 times higher than corn [18]. One hundred and seven active ingredients including acephate, methiocarb, monocrotophos, methomyl, deltamethrin were detected on harvested rose, gerbera and chrysanthemum flowers [29]. Roses were the most contaminated flowers with a total concentration of 26 mg/kg pesticides per a single rose. This increased consumption of pesticides is in part due to the genetic alterations in breeding process of flower varieties in which wild genes that promote their natural defenses were removed [30]. It is also because of the strict international demand for the high quality of flowers including the absence of pathogens [16]. In addition, compared with edible flowers, the lack of determinants of maximum residue limits (MRL) and establishment of criteria for the pesticide application in ornamental flowers also contribute to their indiscriminate use. As estimated by Pereira et al. [16], the greatest absolute frequency of the consumed pesticides were organophosphates, carbamates, triazoles and pyrethroids. The considerable environmental emission of pesticides through leaching and runoff can have huge ecological impacts from environmental disturbance to human health effects. The environment surrounding flower crops can be severely disturbed with the soil and water resources being contaminated and non-targeted organisms being poisoned [18]. About 25% of the total pesticides (chlorothalonil and chlorpyrifos) applied to flower crops in greenhouses can reach the surface soil [31]. The leached pesticides can be strongly adsorbed to the soil and can persist for more than 100 days [32]. Consequently, soil functions can be severely disturbed which can result in overall metabolic rate reduction, soil biodiversity loss and soil fertility decrease [33].

### 2.2. Soil Acidification and Salinization by Abusive Use of Fertilizers

Irrigation and fertilization are particularly important in floriculture as flower crops are demanding for water and nutrients. With 1 kg production of flower, a total consumption of 100–350 kg of water would be expected and a significant portion of water would leak from soil without efficient usage by flower crops [34,35]. Excessive supply of nitrogen, phosphorus, potassium and micronutrients is the norm in the floriculture as essential elements determine the production quantity and quality of flowers [36]. The imbalanced nutrients input and inefficient plant uptake can altogether result in a massive waste of water and nutrients which eventually leaked into the soil ecosystem. For instance, an average of 4.45 g fertilizers were applied to each rose stem in the rose production located in Ethiopia [36]. However, 0.7–1.0 g of fertilizer was already sufficient per rose for its growth and development [37]. Land degradation has been one of the main reasons for the increased fertilizer inputs to maintain soil fertility [38]. Nevertheless, nutrient enrichment by excessive application of fertilizers can diminish the soil function through its impact on soil physicochemical properties and soil biota which in turn lead to further soil degradation such as soil acidification and secondary salinization.

With excessive application of fertilization and a high frequency of irrigation under covered greenhouse conditions in floriculture, the coexistence of soil acidification and salinization frequently occurs [39]. Soil acidification is a natural occurring slow process that can be accelerated by anthropogenic activities like continuous application of nitrogen and sulfur which disturb the nitrogen (N) and carbon (C) cycles [40]. The major cause of soil acidification is the release of H^+^ ions by the strong nitrification of ammonium that exceed the uptake capacity of plants in the N cycle [39]. Acid-neutralizing capacity (ANC) can be reduced when nitrate and base cations (e.g., Ca^2+^, Mg^2+^, K^+^) are leached by the frequent irrigation with H^+^ ions remaining in the soil. Secondary salinization can be also induced by excessive N fertilizers and low recovery rate. Soil is considered salinized when the electrical conductivity (EC) of saturated extract soil water exceeds 4 dS/m, at 25 °C [41]. Secondary soil salinization typically results from an imbalance between transpiration and water inputs from irrigation with soil characteristics that impede or make leaching difficult [42]. Most dissolved salts reside in soil and are apt to accumulate and precipitate at the soil surface when there is a lack of natural precipitation combined with the upward capillary movement of pore water due to evaporation and root uptake [42]. Both soil acidification and salinization pose serious threat to agricultural production by affecting soil function with respect to the nutrient balance by reducing the availability of essential nutrients (e.g., phosphorus, molybdenum, calcium and magnesium) and increasing the mobility and solubility of toxic metals (aluminum and manganese) and salts (boron, chloride, sodium and carbonate). Soil acidification is also a key driving factor in aggravating the occurrence of soil-borne disease [43,44]. Salt precipitation by salinization can greatly affect soil mechanical properties by inducing soil cementation [45].

### 2.3. Persistent Pollutions by Plastic Products

Plastics, as an integral component in agricultural production, provide multiple agronomic benefits in weed and pest control, soil moisture conservation, control soil temperature and enhance nutrient uptake which further translate to increased yield, efficient use of water and nutrient and reduced use of pesticides [46,47]. Plastics are used throughout the life cycle of floriculture from propagation in pre-production stage, flower production and harvest in the production stage, to packaging and transport in the post-harvest handling stage. For example, the consumption of plastic to produce a bunch of rose flowers (20 stems) can be up to approximately 31 g [36]. An estimation of 320 to 408 million pounds of plastic pots, flats, and cell packs were produced annually for the nursery and greenhouse industries [46]. Substantial plastic wastes can enter the soil system due to the over exploitation of plastics and lack of proper waste management in the floriculture. Degradation of these plastics in the soil is limited as polypropylene (PP) after 1-year incubation only had 0.4% degradation rate while 10–35 years soil incubation of polyvinyl chloride (PVC) had no weight loss [48]. Floricultural plastics, composed of conventional polymers such as polyethylene (PE), polypropylene (PP), and polyvinyl chloride (PVC), can physically fragment into micro- and nano-debris and accumulate in large quantities over time in the soil due to their environmental persistence. When these fragments are translocated to adjacent receiving environment and taken up by soil biota, soil functions such as fertility, microbial activity, and biodiversity can be severely hindered which ultimately resulting in crop quality and yield decline. The current status of the recycling and reuse rate for these plastic wastes are still very low (<10%) [49]. Some attempts have been made to provide effective solutions [50] in the decontamination and recycling of plastics container of agrochemicals (e.g., fertilizers and pesticide), which are usually discarded and burnt on the farmland [51]. With the increasing public concerns about the adverse effect of plastic wastes (especially microplastics) to environmental and human health, sustainable use of plastics in floriculture should be seriously considered with regulations from both circular economy principles and soil conservation behaviors.

### 2.4. Flower Wastes: The Elephants in the Room

Not all harvested flowers are qualified for sale on the market. Flower wastes can be generated due to crop maintenance, low qualities, and poor post-harvest handlings in packaging and logistics. For example, monthly waste generation by planting roses can reach to 50 kg per ha [52,53]. In the rose cultivation greenhouse based in Yunnan Province of China, the daily generation of rose stems and flowers can be up to 200 kg/ha (data not published). There was approximately 6% disposal rate of cut flowers during the packaging process due to the low quality [54]. The total amount of floral wastes is alarming in flower-producing developing countries. In Ethiopia, five floriculture farms can generate 66,140 tons of floral waste per year in Batu town [3]. In India, the total amount of flower wastes is approximately 4.74 × 10^6^ t/d [55]. In Yunnan province of China, the daily generation of flower waste in the proximity of KIFA (Kunming International Floral Auction trading center) can reach up to 140 tons (data not published). The common practices of these green waste management include incineration, landfilling, and dumping in open sites, leading to environmental problems such as eutrophication, greenhouse gas emission, and spreading disease [55,56]. Nevertheless, floral wastes are resourceful materials that can be utilized in color extraction, biofuel production, vermicomposting, biochar synthesis and charcoal briquettes [3,57,58]. Instead of wasting these resources, appropriate technology and environmental-friendly management of floral wastes are urgently needed to eliminate the negative impacts of floral wastes on the environment.

## 3. Soil Multifunctionality Loss Due to Disturbance and Contamination in the Floriculture Industry

### 3.1. Disturbed Soil Biota by the Accumulation of Contaminants from Agrochemicals

Soil consists of diverse biologically active organisms that perform many vital functions such as nutrient biogeochemical cycling, soil quality maintenance, and pests and disease regulation. The soil microbial community can decompose recalcitrant polymers into labile monomers that can be readily taken up by plants [59]. Some specific microbes such as N_2_-fixing bacteria, nitrifiers, denitrifiers and P-solubilizers can transform nitrogen and phosphorus into available nutrients [60]. Some plant growth promoting rhizobacteria (PGPR) offer many services to plants such as phytohormone biosynthesis, rhizoremediation of xenobiotic pollutants and biocontrol of pathogens. Burrowing activities by soil organisms such as earthworms can decrease soil erosion and modify soil porosity by increasing aeration, water retention and reducing compaction [61]. Soil detritivores such as nematodes, springtails, pot worms, and millipedes decompose organic materials into available nutrients and increase soil fertility [62]. Many soil fauna such as nematodes, mites and parasitoids also play important roles in controlling pests and weeds [63]. However, with the indiscriminate and successive application of agrochemicals and plastics, accumulating pesticide residues and fragmented plastics in the soil environment can impart multiple negative effects on the soil biota such as fecundity inhibition, growth reduction, food web disruption, and soil biodiversity loss [64,65].

Taking pesticides for an example, a large quantity of the applied pesticides can accumulate and persist in the soil which would reach and cause detrimental effect on the non-targeted soil organisms. Ecotoxicity impact of pesticide in floriculture is particularly greater compared to that of other field crops due to the high pesticide intensity (kg/ha) and ecotoxicity of insecticides and fungicides used, exceeding 20,000 PAF m^3^d/ha (where PAF is the potentially affected fraction of species) [18]. Despite of the huge negative influence from floriculture pesticide, few studies have estimated the effects in the field [66]. Gunstone et al. [67] reviewed 400 studies on the effects of pesticides on non-targeted soil fauna and indicated that pesticide residues pose a clear hazard to soil fauna at both the individual (growth, mortality, behavior, and biomass) and community levels (abundance, richness, and structure etc.). Increased pesticide residues in the soil can also result in microbial metabolism disruption, microbial activity inhibition, beneficial-pathogenic microbes imbalance, microbial diversity loss, and microbial functionary decline, presenting a great risk for the soil sustainability [68,69,70]. Moreover, the frequently applied pre-treatment of soil disinfection by multiple chemical fumigants, which is essential component to control soil-borne pests, can severely diminish the function of microbial consortium that can effectively decompose and utilize pesticide degradation products [70] by indiscriminately killing all living soil organisms. Correspondingly, with specific strains or groups of micro-organisms being significantly inhibited by these biocides, some microbes may acquire resistance against pesticides through single, continuous, or spontaneous mutations trigged by abusive and continuous pesticide use [71,72]. Our previous work revealed that repeated applications of dazomet (50 g per m^2^ over 10 years) in the lilium growing soil would fundamentally affect the microbial composition, with beneficial mycorrhizal species diminished while pathogenic species (i.e., genus *Fusarium*) reassembled, implying the presence of the differential resilience capacity between beneficial microbes and pathogens (Data not published). Consequently, an imbalanced microbial community and disturbed soil fauna community can accelerate the buildup of pathogenic microbes which result in the common problem of continuous cropping obstacles (CCOs) in the intensive flower production systems.

Microplastics (MPs, particle size smaller than 5 mm) and nanoplastics (NPs, particle size smaller than 0.1 μm) that are derived from plastic wastes are another emerging persistent pollutant. This can hinder water infiltration, decrease water holding capacity, impact microbes and macrofauna, and decrease soil fertility, leading to unpredictable deleterious effects on soil function and health [73,74]. Additives such as flame retardants and plasticizers in the plastic products can leach into the soil environment via desorption and degradation, causing toxic effects on soil biota [64]. Nanoplastics can cross the plant cell wall and membrane barriers and transport to other locations via apoplastic pathways [75]. Seed germination, shoot elongation and root growth in crop plants (e.g., cucumbers, wheat, rice and beans) can be severely hindered following the exposure and uptake of plastic debris [76,77]). Plastic particles can disrupt microbiome composition, alter immune response (i.e., immune modulator release, immune cells activation and inflammatory response), change enzyme activity (i.e., catalase, urease, dehydrogenase, alkaline phosphatase, and fluorescein diacetate hydrolase) and interfere gene expression [78,79]. Non-targeted soil fauna (e.g., isopods, nematodes, collembolan, and oligochaeta) can also be adversely affected by plastic debris in growth, survival, metabolism, gut microbiome, feeding patterns and inflammatory reaction [80,81,82,83]. With these increasing disturbances on soil biota and the gradual degradation of soil function, the maintenance of soil sustainability seems challenging to achieve.

### 3.2. Soil Structural Damage, Ion Toxicity and Microbial Imbalance by Soil Acidification and Salinization

Soil salinity often results in salt precipitation that can greatly affect soil mechanical properties. Soil is considered sodic when sodium concentration is high enough to elevate the sodium adsorption ration (SAR) above 13 [41]. Sodic soils usually have a cloddy structure, as soil precipitation can induce soil cementation that holds sand and silt particles together [45]. Salt precipitation in the soil pores can reduce soil aggregate stability, percolation and evaporation processes and it can increase surface runoff as the crystalized salts reduces matrix porosity, permeability and vapour diffusivity [41]. Thus, high soil salinity has critical impacts on soil-water relations leading to erosion, low field capacity, and plant inhibition [84].

Both soil acidity and salinity can cause yield loss due to ion toxicity, nutrient deficiency, and osmotic stress. In acid soils, the higher H^+^ concentration increases the mobility and solubility of toxic metals such as aluminum (Al^3+^) and manganese (Mn^2+^). Al ion becomes hazardous upon interactions with plant sub-cellular organelles and induces various detrimental effects including inhibited root growth, reduced water absorption, phosphorus deficiency, oxidative bursts and DNA denaturation [85]. Excessive Mn can also adversely affect plants in uptake and translocation of essential nutrients (e.g., Ca, Mg, and P), chlorophyll biosynthesis, and root development [86]. On the other hand, soil acidity can enhance the retention of phosphorus through adsorption and precipitation with Fe and Al [87]. Deficiency of base cations such as Ca^2+^ and Mg^2+^ can be also induced due to their leaching of exchangeable forms in acidic soils [88]. Moreover, the residence time and leaching potential of toxic metals (Cd and As) can be increased as soil pH decreased which consequently affects plant growth [89]. High H^+^ influx in roots due to low soil pH also adversely affect the root tissue by membrane depolarization and cytoplasm acidification, which causes substantial reduction in plant growth and yield [90]. A decreased crop yield can be also observed if soil salinity exceeds crop-specific thresholds. Sodic soil hinders root penetration and reduces water and oxygen uptake due to its disperse nature and loss of soil pores [84]. This inability to access water impedes osmoregulation and induces plant wilt [91]. High levels of soil sodium can lead to potassium deficiencies due to K^+^ transporters cannot distinguish between the two ions (Na^+^ and K^+^) upon their similarities, leading to sodium toxicity in leaf scorching and bronzing [92,93]. Salinity inhibits cellular homeostasis and denatures important biological molecules, subsequently negatively affecting crop yield and quality by hindering photosynthesis, pollen sterility, and seedling vitality [94,95]. Chloride toxicity is also common in saline soils as Cl^−^ does not adsorb onto negatively charged soil particles and moves freely with soil water. Anion channels in plant roots allow Cl^−^ to passively move into root cells which can interfere with photosynthesis leading to chlorosis and leaf burn and necrosis [96].

High soil acidity and salinity can have significant impacts on soil biodiversity and associated ecological functions. Soil microbial activity and community structure can be regulated by soil salinity due to selective pressure from osmotic stress [97,98]. Several studies indicated that fungal community is more tolerant to salinity stress compared to bacteria due to the protection provided by their chitinous cell walls against low matric potentials [99]. Salinity effect on the shift of soil bacterial community is also attributed to overall high community salt tolerance [98]. These alterations in microbial community by increasing salinity often result in the decreased soil enzymatic activities involved in decomposition of organic matter and nutrient cycling [100,101,102]. Soil acidification has a direct influence on microbial diversity by suppressing beneficial microbes (e.g., *Bacillales*, *Burkholderiales*, and *Pseudomonadales*) and enriching plant pathogens (e.g., *Fusarium oxysporum*, *Ralstonia solanacearum*, and *Plasmodiophora brassicae*) [43,103,104]. Meta-analysis by Zhang et al. [44] demonstrated that the decrease of soil pH value was the key driver of soil-borne disease occurrence. The authors suggested that it could be attributed to their varying capability of pH tolerance as fungi possess a generally broader range (2.0–13.0), while bacteria with N_2_ fixation and nitrification abilities do not develop when soil pH is below 4.5. The synergistic interactions such as nodulation and N fixation amid plants and linked rhizobia can thus be adversely impacted [105]. Floricultural crops are often sensitive to salinity and acidity which required a relatively narrow range in which to develop and grow [106,107]. Thus, only by protecting soil from the irrational use of fertilizers and irrigation can guarantee the quality of flower production and maintain the sustainability of soil function.

### 3.3. Continuous Cropping Obstacles in Floriculture: Balance Between Economic and Ecological Benefits

Soil can be modified by the legacy of previous abiotic and biotic processes and current living plants, and soil biota can respond to this modified soil condition in real time while again altering the soil conditions for subsequent plants and soil biota [108,109]. This loop is often described as plant-soil feedback (PSF) [110], which can be considered as niche reconstruction by modifying or destroying their own niches or the niches of other organisms by interaction networks among plant, soil biota and soil [111]. Several pathways can be distinguished in the plant-soil feedback loop (illustrated in Figure 2), such as the biota pathway consists of plant-soil biota interactions, the soil pathway encompassing plant-soil interactions, and the biota-soil pathway consists of plant-soil biota-soil interactions [109]. A classic example of plant-soil biota pathway is the facilitative interaction between plant allelopathy autotoxins (via root exudates or litter leachates) and soil pathogenic microbes. For example, autotoxins emitted by Lanzhou lily (*Lilium davidii* var. *unicolor* Cotton) through root exudates such as phthalic acid, 4-vinyl guaiacol and 2,4-di-tert-butylphenol can enhance the sporulation and pathogenicity of *Fusarium oxysporum* by stimulating the synthesis of mycotoxins and pathogenesis-related hydrolytic enzyme, which can ultimately promote CCOs [112,113]. In the soil pathway, negative PSF is induced by altering soil physicochemical properties. For instance, the disrupted soil N and S metabolism, nutrient imbalance (i.e., available N depletion) and increased soil salinization contributed to the CCOs of cut chrysanthemums [114]. The biota-soil pathway is more complex with tri-partite interactions among plants, soil biota and the soil environment. Our previous study on the CCO of lisianthus plant (*Eustoma grandiflorum*) revealed that soil salinization, beneficial microbe loss and phyto-autotoxin accumulation along with their interactions all contribute to the soil function degradation and CCO build-up in the field [115]. These pathways with plants playing the central role can appear immediately or cause accumulative alterations, which are all preserved in the soil over time and cause long-term legacy [109]. Nevertheless, standard and persistent agronomic practices in floriculture such as repeated use of the same agrochemicals, irrigation and harvest approaches, tillage depth, and farm machinery make this PSF loop more human-dominated.

Anthropogenic activity is the main driving force shaping the structure and function of agro-horticultural ecosystem, including floriculture [116]. Due to the disturbance of agrochemicals and plastics in the soil as aforementioned, the prominent CCOs in floriculture are inevitable. As classic negative plant-soil feedback, CCOs can entail substantial economic losses in yield penalties and compromise the sustainability of floriculture in overall soil degradation [117,118]. The means to overcome CCOs depend on the underlying mechanisms, which vary greatly among different flower species. Given that CCOs in floriculture is mostly attributed to the intensive inputs involving huge human investments, it is thus of great significance to increase the ecological awareness of sustainability among stakeholders as perception of eco-friendly agronomic practices and adoption attitudes are the most important factors influencing adoption of sustainable decisions [119]. Sustainable development is not only about maintaining environmental integrity, but also requires promoting economic prosperity and social equity as the primacy of corporate financial performance over environmental and social concerns is prominent in floriculture [120,121]. Therefore, exploration and implementation of technologies with both economic and ecological benefits that are suitable to overcome the decline in soil health and improve the poor perception of sustainable floriculture concept among stakeholders are necessary. Other than that, efforts from government in executing and improving subsidy levels when popularizing novel technologies and establishing eco-friendly floricultural products are also crucial to provoke farmers’ and companies’ perspective of sustainability through customer demand [119,122].

## 4. Sustainable Practices in Preserving Soil Ecological Functions in Floriculture

Multiple practices are listed below that can be implemented to achieve soil health in the development of floriculture sustainability, from cultivar breeding to recycling of flower wastes (illustrated in Figure 3).

### 4.1. Breeding of Stress Resistant and Nutrient Efficient Varieties

Stress resistant and nutrient efficient flower cultivars require fewer pesticides and fertilizers, making the production process more sustainable. Nevertheless, the current breeding goals of flower cultivars are primarily focused on their ornamental characteristics such as color, fragrance, plant architecture, and post-harvest longevity [8,130,131,132]). In addition, selective breeding for those decorative phenotypic traits often leads to poor resistance against various stresses when compared to their wild counterparts [133,134], which increases the usage of pesticides. With the growing awareness in the sustainability of floriculture, however, shifts in legislation will eventually affect the conventional production modes. In Europe, the publication of the European Green Deal sets concrete targets in reducing at least 50% of the pesticide use and 20% of the fertilizers use by 2030 [135]. This will require growers and breeders to pay much more attention to sustainable pathways and practices, including setting new breeding goals in enhanced stress resistance and increased nutrient efficiency [128].

Classic breeding strategies (such as intra- and inter-specific hybridization, mutagenesis, polyploidization, and double haploid induction, etc.) will not suffice in achieving this goal due to the evident limitations and drawbacks such as high heterozygosity, long life cycle, and self-sterility [136,137]. Furthermore, conventional breeding in ornamental plants has other major limitations such as complex genomes, limited availability of gene pools, and a lack of genetic variability [138]. With the advances in next-generation sequencing (NGS) and multi-omics technologies, an enormous resource platform with complete genome information can be established to facilitate the revolution of breeding technology such as genetic engineering technology and transgenics [139,140,141,142]. This can facilitate significant advances in ornamental crops with improved traits such as increased resistance to pests and diseases and more sustainable production practices [143,144,145]. Many efforts have been made through genetic manipulation in ornamental plants such as rose, lily, gerbera, chrysanthemum, tulip, and carnation to strengthen their responses against abiotic and biotic stresses [144,146,147]. For instance, many responsive genes involving in plant defense to fungal infection are introduced into roses, chrysanthemums and carnations to enhance their resistance to multiple fungal pathogens such as powdery mildew, black spot, leaf spot, gray mold, and Fusarium wilt [138]. Important genes with potential functions in abiotic stress tolerance (e.g., drought, cold, salinity, and heat stress) have also been intensively investigated in ornamental crops for breeding purposes [147,148]. However, nutrient use efficiency has traditionally only been a target in food crops such as rice, wheat and maize [149,150,151]. In order to develop the sustainability of floriculture, breeding nutrient efficient ornamental cultivars that helps in optimizing nutrient acquisition (i.e., the amount of nutrients that taken up by plants in relation to the supply) and nutrient utilization (i.e., the biomass produced by the unit of nutrient incorporated by plants) should be considered in future breeding targets [152].

### 4.2. Sustainable Soil Management: A Long Way to Go

Floriculture sustainability can be increased through sustainable and resilient agricultural practices that improve productivity, strengthen soil capacity, maintain soil function, and progressively upgrade land and soil quality. Soil is at the core of the sustainability paradigm for human development [28]. Therefore, sustainable soil management is necessary to break the negative spiral of soil degradation and environmental damage [153], seeking at the same time to make floriculture sustainable. The concept of floriculture sustainability should align with agricultural sustainability which considers the environmental, economic, and social aspects of farming, while also promoting the resilience and persistence of productive farmlands [154]. One of the forward-looking solutions is the adoption of site-specific soil management integrating nutrient, water and pest management practices that can improve soil productivity, the efficiency of external inputs, and farm outputs [123,124,125,126,127].

Sustainable soil management involves the use of modern technologies and traditional methods, and it encompasses site-specific and continuous improvement across the whole farm [155]. Modern novel agronomic approaches include inoculation of beneficial microbes such as PGPR (plant-growth promoting rhizobacteria), the application of biocontrol agents based on ecological interactions such as predatory mites, efficient resource recycling of agro-industrial wastes such as composting and green manuring, and computerized monitoring and analytic systems such as decision support systems and models for fertigation [8,23,156,157]. Traditional methods are based on “informal” knowledges gained from cultural knowledge and work experience such as diversified cropping such as intercropping, tied ridges, and use of organic fertilizers [125]. Sustainable soil management requires interdisciplinary interaction of natural and socio-economic sciences to achieve a systematic understanding of local situations (e.g., local pedo-climatic and socio-economic conditions), to conceptualize and contextualize local problems (e.g., specific shortage of certain soil nutrients or outbreak of certain pests), and to make scientific evidence useful for practical implementation (e.g., site-specific nutrient, irrigation and pest management) [25]. Nevertheless, a prominent problem with current sustainable soil management is the knowledge fragmentation across disciplines and weak awareness among agricultural practitioners (i.e., farmers, advisors, industry representatives and policymakers) [25,155]. In addition, existing soil management technologies are often underdeveloped, insufficiently tested, or lack proper marketing, competitiveness and legal support which helps encouraging users towards soil protection or providing nature-based solutions [25]. Taking integrative pesticide management that has been developed for 6 decades as an example, several challenges still remain to be remedied such as unnecessary confusion of the concept, inconsistencies between the concept and its actual practice, the unguided engagement of farmers, inadequate research and insufficient consideration of ecology [124]. For sustainable soil management to be adopted in full potential in floriculture on a systematic level, it needs to maintain a programmatic overview of all related challenges and interdisciplinary knowledge types and to balance tradeoffs together with stakeholders (those affected by soil management) and actors (those engaged with soil management) to include their opinions, perspectives, problems, possibilities and realities [158]. Considering the complexity of socio-economic interrelations with soil health and related policies for practical application of sustainable soil management in floriculture, we still have a long way to go.

### 4.3. Value-Added Products from Flower Wastes: Fallen Blossoms Are Not an Unfeeling Thing

Recycling flower wastes within the production cycle contributes to the sustainable development of the floriculture industry by lowering the negative environmental impacts [3,52,55,58,129]. Unfortunately, environment-friendly disposal of flower wastes is often difficult to achieve due to the lack of understanding of the nuances of issue by stakeholders, inadequate environmental legislations and industry standards, and insufficient academic research upon which to base regulations. Flower waste is easy to collect from its source of generation (i.e., cultivation bases and trading markets) without the need to separate it into various categories in comparison with other green wastes [55]. Several biotechnological approaches have been evaluated for reliable methods of flower refuse management including composting, vermicomposting, dye extraction, biochar synthesis, biogas generation and other options [55]. For example, products from co-composting rose wastes is a good organic fertilizer with high nutrient contents, low polyphenol levels and disease suppressive potentials [52]. Singh et al. [58] converted the marigold flower (*Tagetus erecta* L.) into biochar at 350 °C and 500 °C respectively which can act as potential adsorbent material for soil amelioration. The charcoal briquettes derived from flower wastes generated by a local floriculture farm had a higher thermal efficiency than commonly used acacia charcoal [3]. Nevertheless, few studies have evaluated these approaches from a holistic management point of view. Establishment of cost-effective flower waste management practice with the potential of large-scale implementation with flower cultivation is still challenging but is urgently needed. Taking composting/vermicomposting for an example, the final products are highly influenced by environmental variables, nature of the composting material, and the growth and activities of microbes and earthworms [129]. Furthermore, the presence of pesticide residues in these flower wastes is inevitable which is mostly ignored during the composting without the pesticide degradation analysis [52].

Biochar is a solid charcoal-like material produced by the thermal conversion of organic biomass using pyrolysis, torrefaction or gasification technologies [159]. Biochar can assist plant-related industries in achieving productivity and sustainability through multiple ways. Converting green organic wastes into biochar helps reducing the negative environmental pollution [160,161,162], GHG (greenhouse gas) emission [163,164] and the spread of plant diseases [165,166]. Biochar substitutes growth media (e.g., peat) and amendments into soil can improve nutrient and water retention [167], enhance soil physiochemical properties [168], enhance the activity carbon sequestration [169], and thus increase plant production and profits [159,170,171]. Our previous work on transferring flower wastes (i.e., rose, chrysanthemum, lisianthus, lily, and carnation) into biochar using superheated steam (SHS) torrefaction [172,173] revealed that flower waste biochar contains higher levels of P (reaching up to 8.75 mg/kg), macropores that suitable for microbial colonization, and microdose (ng/mg) of various polyphenols (See Table 1). Specifically, 1–3% addition of biochar from rose wastes in the growth substrate can significantly stimulate the growth of lettuce from both photosynthesis activity to root vitality (See Figure 4). However, great efforts are still needed to demystify the application possibility of floral waste biochar into flower cultivation along with the underlying mechanisms of growth stimuli and disease suppression.

### 4.4. “Reuse and Recycle” over “Disposal After Use” in Plastics

Sustainable use of plastics in floriculture should align with the “3R” waste hierarchy concept as reducing, reusing and recycling. Sustainable treatment options of plastics depend on their collectability after use and their life span [46]. Polymers that do not chemically weather and fragment are preferred for plastics that allow complete collection after use such as agrochemical containers and greenhouse covering. For those plastics that is difficult to recycle such as mulch films or coatings for controlled-release fertilizers, biodegradable plastics that is less toxic are preferred over conventional persistent polymers. Problematic plastic types such as oxo-degradable mulches, polyvinyl chloride and other microplastics should be selectively banned. Agricultural plastics are in general recyclable from greenhouse plastic films to plastic bale wrap [174]. Plastics can be recycled by mechanical and chemical processes such as shredding, re-extrusion, pyrolysis, dissolution, enzymatic methanolysis, and gasification [175,176]. Recycled plastic is highly used in manufacturing industries for cost reduction and to produce various material such as aromatic olefins, plastic resins and original plastic monomers, methanol, syngas, and transportation fuels [177]. For those plastics that cannot be collected and recycled such as thin mulch films, geotextiles, and abraded plastic from string trimmers and plant clips, it is necessary to consider plastic biodegradability to ensure no residues accumulations in the soil [46]. Nevertheless, the environmental capacity of biodegradable plastics requires more realistic filed conditions over long period instead of laboratory incubation tests [178]. Biodegradability standards can thus be defined based on the academic results to specify the test condition and analytical methods [179]). However, biodegradable plastics are more expensive to produce than conventional plastics, which may considerably reduce its acceptability [46,179]. Global growth for bioplastics could reach up to 10–20% if political support or subsidy can be realized [180]. To ensure the “reuse and recycle” of floriculture plastics, greater efforts from policymakers and companies are required as action plans are only feasible either from legislation and regulations or voluntary initiatives. For example, the high growth rate for bioplastics in Europe is driven by both recycling-oriented regulations and increased consumer demand for sustainable plastic products [181]. Meanwhile, collection facilities that are well-coordinated and widely accessible are also essential to facilitate plastic recycling [46].

## 5. Concluding Remarks and Future Perspective

The floriculture industry is experiencing rapid growth and profitability. It also faces significant challenges that can impede its sustainable development. One of the most pressing issues is the environmental degradation caused by the production practices in floriculture, particularly its impact on soil, which is the fundamental element to agriculture. This study addresses various environmental concerns related to soil health in modern floriculture and outlines strategies for achieving sustainability.

Developing and implementing sustainable soil practices in floriculture is a complex task, as research, practice communities, and policy often operate in isolation with limited collaboration. A holistic overview of the problems related to floriculture soil and the integrated involvement of all stakeholders are thus crucial. Bridging the understanding gap and raising awareness of the often-overlooked soil threats and their consequences among growers, companies, and policymakers is of utmost importance. Soil sustainability issues in the flower industry are not solely the result of actions by growers and companies but also stem from the absence of science-based policies at multiple governmental levels. Coherent and effective policies, including sustainable soil management guidelines and specific requirements, are important for the practical implementation of sustainable technologies and methods among growers and companies. For instance, subsidies support linked to good practice and actual soil monitoring combined with penalties for damaging soil functions or the introduction and popularization of novel environmentally-friendly agrochemicals help with navigating individual and industrial activities towards more sustainable soil managements. By increasing awareness of soil degradation and multifunctionality, along with greater engagement of participants and stakeholders, can accelerate the transition towards sustainable floriculture.

The academic community in floriculture serves as a hub for knowledge creation, fostering education, promoting novel technologies, and encouraging practical implementations. However, a major issue within current floriculture research is the negligible consideration of the ecological consequences of pre-production and production management, with a primary focus on breeding new varieties. Consequently, knowledge regarding the prevention of soil damage and the protection of soil function in floriculture is severely lacking, impeding the sustainable management of soil that could guide growers or policymakers in adjusting their understandings, behaviors, and decisions. To address this deficiency, it is imperative to involve soil ecologists or soil scientists with experience in agricultural soil to establish a systematic view of the identified problems and to highlight all relevant aspects within floriculture (preferably through stakeholder engagement). Based on this background investigation, scientists can conduct specific experiments to collect data and evidence demonstrating the threats or employ innovative techniques to illustrate the benefits. Sustainable soil management can thus be supported and further developed through field observations and interactions with stakeholders. The acquired knowledge can then support the practical applications and implementations of innovative methods by stakeholders to achieve floriculture sustainability. Nonetheless, support from governmental funding and investment in soil research and in the development of sustainable practices within the floriculture academic community remains the most significant bottleneck.

## Figures and Tables

**Figure 1 plants-13-03055-f001:**
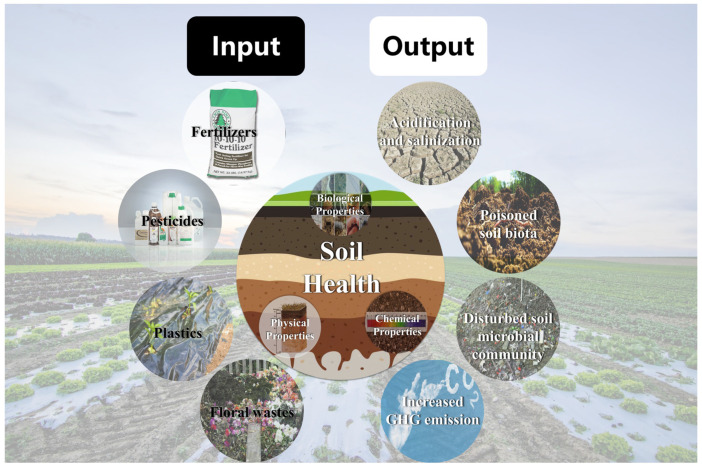
A summary of soil threats from soil acidification and salinization, soil biota poisoning and soil microbiota disturbance, and increased GHG emission by the excessive use of fertilizers, pesticides, and plastics along with the poor treatment of floral wastes.

**Figure 2 plants-13-03055-f002:**
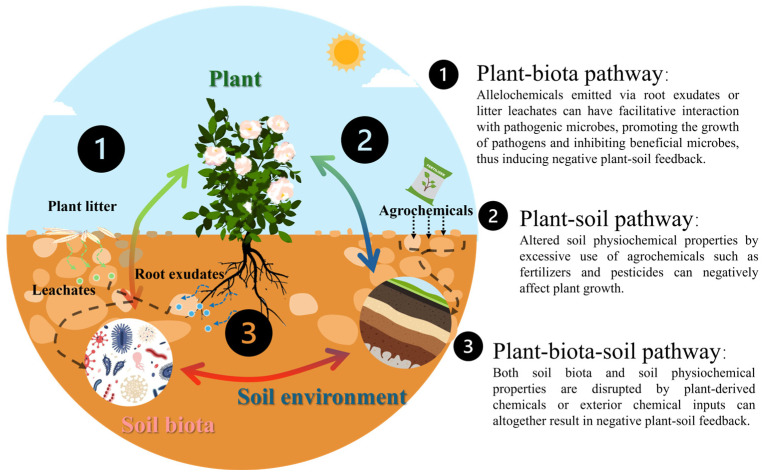
Three pathways of plant-soil feedback based on the description of Frouz [109]: 1. Plant-biota pathway; 2. plant-soil pathway; and 3. plant-biota-soil pathway. Dashed line indicates the direct effect from root exudates or leachates to soil biota and agrochemicals to soil physiochemical properties. The solid line indicates the interactions among plant, soil biota, and soil environment.

**Figure 3 plants-13-03055-f003:**
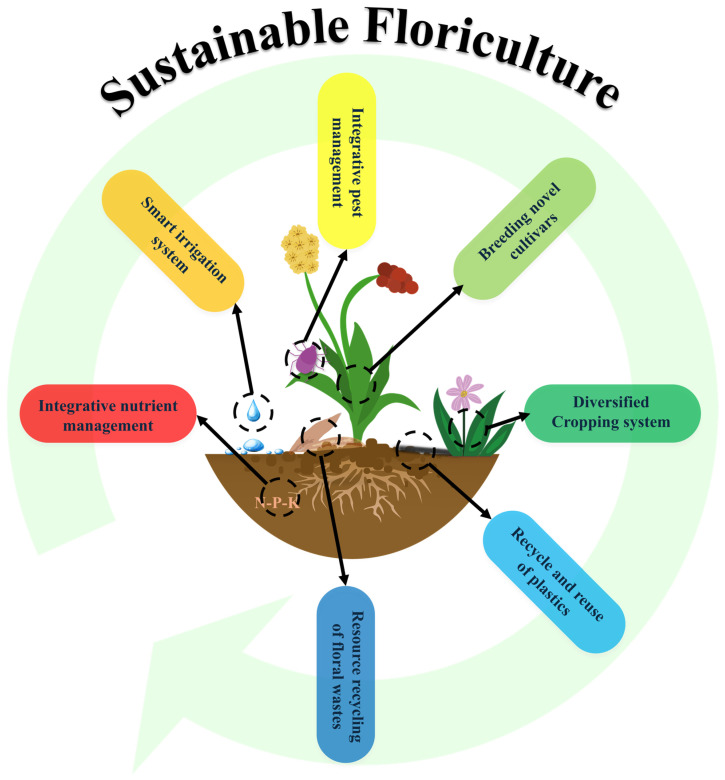
Multiple agronomic approached that can be used to achieve soil health in the development of floriculture development, including integrative nutrient, water and pest management [123,124,125,126,127]; breeding cultivars with breeding goals in enhanced stress resistance and increased nutrient efficiency [128]; suitable traditional methods such as diversified cropping system and use of organic fertilizers [125]; sustainable use of floriculture plastics [46]; and resource recycling of flower wastes such as biochar synthesis, composting and charcoal generation [3,52,55,58,129].

**Figure 4 plants-13-03055-f004:**
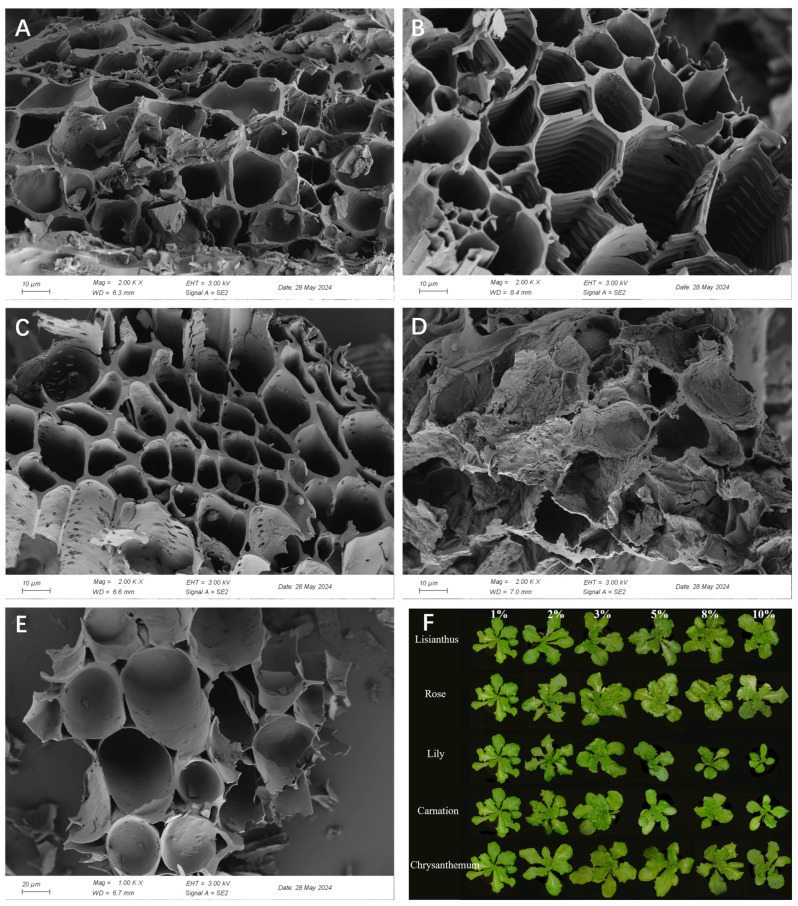
Scanning Electron Micrograph of biochar derived from floral wastes (**A**) rose, (**B**) lily, (**C**) lisianthus, (**D**) chrysanthemum, (**E**) carnation, and (**F**) the effects of lettuce growth with their application (1,2,3,5,8,10% addition) in growth substrate.

**Table 1 plants-13-03055-t001:** Basic physicochemical properties of biochar derived from flower wastes using superheated steam (SHS) torrefaction.

Flower Type	Phosphorus (mg/g)	Polyphenols (ng/mg)	CEC (mol/kg)	Macropore (μm)
Rose	5.81	515.84	34.27	>10
Lily	6.63	479.73	48.03	>20
Lisianthus	3.01	304.73	30.7	>10
Chrysanthemum	8.75	518.09	41.2	>10
Carnation	7.47	457.31	40.13	>20

## Data Availability

Data available on request. The data underlying this article will be shared on reasonable request to the corresponding author.

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
