# Peer review of "From Flourish to Nourish: Cultivating Soil Health for Sustainable Floriculture"

_plants, 2024, doi:10.3390/plants13213055_

Round 1
Reviewer 1 Report
Comments and Suggestions for Authors
The review paper ''From Flourish to Nourish: Cultivating Soil Health 3 for Sustainable Floriculture'' is clearly written and well organized. In the Introduction section the importance of floriculture industry is explained, as well as its environmental impact due to excessive inputs of agrochemicals and energy consumption. The influence of intensive production is well analyzed with enough data cited to explain the influence on soil in regards with overexploitation of soil resources and increase of soil disturbance. The third chapter 'Soil multifunctionality loss due to disturbance and contamination in the floriculture industry'' is clearly written with good examples and explanations. The fourth chapter presents examples of sustainable practices in preserving soil ecological functions which can be used as a basis for future research and organizing the sustainable floriculture production.
This manuscript contains more than 170 references which are properly cited in text. The authors have experience in this filed of research, according to their other published research papers. There are no similar papers presenting this evidence. The only similar review regarding soil health and agriculture is published in 2020 (M. Tahat, M.; M. Alananbeh, K.; A. Othman, Y.; I. Leskovar, D. Soil Health and Sustainable Agriculture. Sustainability 2020, 12, 4859. https://doi.org/10.3390/su12124859), but this paper has different approach and different data were analyzed.
The review paper ''From Flourish to Nourish: Cultivating Soil Health 3 for Sustainable Floriculture'' is well written, with lot of data clearly presented, more than 170 references is cited in this paper. I recommend this paper to be accepted for publishing. There are no suggestions for changing text.
Author Response
Comments1:
The review paper ''From Flourish to Nourish: Cultivating Soil Health 3 for Sustainable Floriculture'' is clearly written and well organized. In the Introduction section the importance of floriculture industry is explained, as well as its environmental impact due to excessive inputs of agrochemicals and energy consumption. The influence of intensive production is well analyzed with enough data cited to explain the influence on soil in regards with overexploitation of soil resources and increase of soil disturbance. The third chapter 'Soil multifunctionality loss due to disturbance and contamination in the floriculture industry'' is clearly written with good examples and explanations. The fourth chapter presents examples of sustainable practices in preserving soil ecological functions which can be used as a basis for future research and organizing the sustainable floriculture production.
This manuscript contains more than 170 references which are properly cited in text. The authors have experience in this filed of research, according to their other published research papers. There are no similar papers presenting this evidence. The only similar review regarding soil health and agriculture is published in 2020 (M. Tahat, M.; M. Alananbeh, K.; A. Othman, Y.; I. Leskovar, D. Soil Health and Sustainable Agriculture. Sustainability 2020, 12, 4859. https://doi.org/10.3390/su12124859), but this paper has different approach and different data were analyzed.
The review paper ''From Flourish to Nourish: Cultivating Soil Health 3 for Sustainable Floriculture'' is well written, with lot of data clearly presented, more than 170 references is cited in this paper. I recommend this paper to be accepted for publishing. There are no suggestions for changing text.
Response1: We thank the reviewer for his or her positive evalution of this review. The manuscript has been revised according to the comments from other reviewers. We hope the revised manuscript can be better for future publication.

Reviewer 2 Report
Comments and Suggestions for Authors
I have reviewed the manuscript and have the following suggestions: The manuscript draws attention to the dangers of soil use in floriculture. I think this is a very important topic that is missing - very little is written about it, and it is becoming increasingly important in the system of sustainable agriculture. The title, abstract and Introduction chapter are adequate, but the wording of the objective and hypothesis is vague and unclear. I suggest rewording and making it more concrete. The literature used is in order. Figures and tables are coherent in the text and informative. I accept the manuscript after minor revisions and recommend it for publication.
Author Response
Comments1: I have reviewed the manuscript and have the following suggestions: The manuscript draws attention to the dangers of soil use in floriculture. I think this is a very important topic that is missing - very little is written about it, and it is becoming increasingly important in the system of sustainable agriculture. The title, abstract and Introduction chapter are adequate, but the wording of the objective and hypothesis is vague and unclear. I suggest rewording and making it more concrete. The literature used is in order. Figures and tables are coherent in the text and informative. I accept the manuscript after minor revisions and recommend it for publication.
Response1: We thank the reviewer for his or her positive evaluation on our work. We have rephrased the objective and hypothesis as suggested by the reviewer. Please see Line 167-175 for the new changes.

Reviewer 3 Report
Comments and Suggestions for Authors
From Flourish to Nourish: Cultivating Soil Health for Sustainable Floriculture
Peihua Zhang1,2,3,*, Jie Zhou4, Di He4, Yiran Yang4, Zhenhong Lu1, Chunmei Yang1,
Dongdong Zhang5, Fan Li1,2, *, Jihua Wang1, 2,*
This is an excellent paper that is well presented. The literature review and discussion sections of the paper are of high quality. While it is well written, there is still a moderate amount of revision and editing needed prior to publication. I recommend its acceptance and publication following revisions. I have provided some direct editorial recommendations (highlighted in red) throughout the narrative and references sections for the authors to consider in their revision. There is a tendency to have long, run-on sentences that need to be broken up, and I have provided some additional wording changes for the authors consideration. Throughout the manuscript I have provided highlighted issues and potential wording changes to address them. I hope that these recommendations are helpful to authors as they revise the manuscript for publication.
References:
Nearly all (but not all) of the references are cited in the narrative/text, and I have highlighted (in red) a few that do not appear to be cited, so that citations can be added and incorporated into the narrative. In a few references the DOI does not appear to function, and that issue can be amended by inserting a replacement DOI - or the DOI can be omitted. In several sets of references there needs to be additional citation information added – such as adding an lower case letter such as a, b or c after the date to indicate which of several publications by an author is being referred to in the narrative (2024, 2024a, 2024b, and so forth). Scientific names should be italicized throughout the manuscript, including in the references (I italicized those I noticed and highlighted them in red).
The authors might consider including these papers in their review of the topic:
Kabir, E., Kim, Ki-Hyun, Kwon, E.E., 2023. Biochar as a tool for the improvement of soil and environment. Front. Environ. Sci., Sec. Toxicology, Pollution and the Environment. Volume II.
https://doi.org/10.3389/fenvs.2023.1324533 This article should be considered for inclusion and citation in the narrative
Olubusoye, B.S., Cizdziel, J.V., Wontor, K., Grandberry, T., Bennett, E.R., 2024. Removal of microplastics from agricultural runoff using biochar: a column feasibility study. Frontiers in Environmental Science 12. https://doi.org/10.3389/fenvs.2024.1388606 This article should be considered for inclusion and citation in the narrative

Author Response
Comments1:
This is an excellent paper that is well presented. The literature review and discussion sections of the paper are of high quality. While it is well written, there is still a moderate amount of revision and editing needed prior to publication. I recommend its acceptance and publication following revisions. I have provided some direct editorial recommendations (highlighted in red) throughout the narrative and references sections for the authors to consider in their revision. There is a tendency to have long, run-on sentences that need to be broken up, and I have provided some additional wording changes for the authors consideration. Throughout the manuscript I have provided highlighted issues and potential wording changes to address them. I hope that these recommendations are helpful to authors as they revise the manuscript for publication.
Response1: We thank the reviewer for his or her positive evaluation and valuable suggestions on this manuscript. We have carefully revised the manuscript accordingly. The long sentences have been rephrased as suggested by the reviewer. We hope these changes can meet the standards for the publication. Please see the changes throughout the manuscript in highlighted parts.
Comment2: References: Nearly all (but not all) of the references are cited in the narrative/text, and I have highlighted (in red) a few that do not appear to be cited, so that citations can be added and incorporated into the narrative. In a few references the DOI does not appear to function, and that issue can be amended by inserting a replacement DOI - or the DOI can be omitted. In several sets of references there needs to be additional citation information added – such as adding an lower case letter such as a, b or c after the date to indicate which of several publications by an author is being referred to in the narrative (2024, 2024a, 2024b, and so forth). Scientific names should be italicized throughout the manuscript, including in the references (I italicized those I noticed and highlighted them in red).
The authors might consider including these papers in their review of the topic:
Kabir, E., Kim, Ki-Hyun, Kwon, E.E., 2023. Biochar as a tool for the improvement of soil and environment. Front. Environ. Sci., Sec. Toxicology, Pollution and the Environment. Volume II.
https://doi.org/10.3389/fenvs.2023.1324533 This article should be considered for inclusion and citation in the narrative
Olubusoye, B.S., Cizdziel, J.V., Wontor, K., Grandberry, T., Bennett, E.R., 2024. Removal of microplastics from agricultural runoff using biochar: a column feasibility study. Frontiers in Environmental Science 12. https://doi.org/10.3389/fenvs.2024.1388606 This article should be considered for inclusion and citation in the narrative
Response2: We apologize for the mistakes made in the reference and citation styles and thank the reviewer for his kind reminder of these mistakes. We have carefully revised the manuscript and made changes in the citation style with additional citation information being added or mistakes highlighted by the reviewer. The recommended literatures were included in the manuscript in the floral waste part. We hope the revised mauscript can meet the standard of publication. Please see changes throughout the manuscript in the highlighted parts.
